# Decision Support Tools for Coronary Artery Calcium Scoring in the Primary Prevention of Cardiovascular Disease Do Not Meet Health Literacy Needs: A Systematic Environmental Scan and Evaluation

**DOI:** 10.3390/ijerph191811705

**Published:** 2022-09-16

**Authors:** Shannon McKinn, Carys Batcup, Samuel Cornell, Natasha Freeman, Jenny Doust, Katy J. L. Bell, Gemma A. Figtree, Carissa Bonner

**Affiliations:** 1Sydney School of Public Health, Faculty of Medicine and Health, University of Sydney, Sydney 2006, Australia; 2Australian Women and Girls’ Health Research Centre, School of Public Health, University of Queensland, Brisbane 4006, Australia; 3Kolling Institute, University of Sydney, St Leonards 2065, Australia

**Keywords:** decision making, shared, patient education, cardiovascular diseases, primary prevention, health literacy

## Abstract

A shared decision-making approach is considered optimal in primary cardiovascular disease (CVD) prevention. Evidence-based patient decision aids can facilitate this but do not always meet patients’ health literacy needs. Coronary artery calcium (CAC) scans are increasingly used in addition to traditional cardiovascular risk scores, but the availability of high-quality decision aids to support shared decision-making is unknown. We used an environmental scan methodology to review decision support for CAC scans and assess their suitability for patients with varying health literacy. We systematically searched for freely available web-based decision support tools that included information about CAC scans for primary CVD prevention and were aimed at the public. Eligible materials were independently evaluated using validated tools to assess qualification as a decision aid, understandability, actionability, and readability. We identified 13 eligible materials. Of those, only one qualified as a decision aid, and one item presented quantitative information about the potential harms of CAC scans. None presented quantitative information about both benefits and harms of CAC scans. Mean understandability was 68%, and actionability was 48%. Mean readability (12.8) was much higher than the recommended grade 8 level. Terms used for CAC scans were highly variable. Current materials available to people considering a CAC scan do not meet the criteria to enable informed decision-making, nor do they meet the health literacy needs of the general population. Clinical guidelines, including CAC scans for primary prevention, must be supported by best practice decision aids to support decision-making.

## 1. Introduction

A coronary artery calcium (CAC) scan is a computerised tomography (CT) scan of the heart to identify calcification occurring in coronary atherosclerotic plaque. A CAC standardised score (based on the area and the density of calcium in the coronary arteries) is increasingly used when deciding about starting blood pressure- and lipid-lowering medication. As no contrast is given, it is only the calcified plaque that is visualised, with the potential for missing more vulnerable non-calcified plaque. However, in a screening setting of a stable and asymptomatic patient, this is strongly associated with subclinical disease burden and risk of events [1]. Specifically, a positive CAC score, measured in Agatston units (AU), reflects the burden of atherosclerosis, with increasing CAC scores correlating to both total (calcified and non-calcified) plaque burden, as well as increasing risk of cardiovascular disease (CVD) events. A score of 0 AU indicates an absence of CAC and a very low (but not zero) risk of CVD events in the next 10 years [1].

The American Heart Association and European Society of Cardiology guidelines both recommend the targeted use of CAC scans for CVD risk screening [2,3], but the US Preventive Services Task Force has stated that the evidence is insufficient to recommend its use [4]. The role of CAC in primary prevention remains controversial due to the lack of direct evidence on its effectiveness in reducing CVD outcomes and limited evidence of clinical utility beyond that provided by current risk assessment tools [4,5,6]. Despite its strong prognostic ability as a stand-alone test, studies have found only modest prognostic value beyond traditional risk factors [7]. Adding CAC into CVD risk assessment also raises issues about cost-effectiveness and equity of access [4,8]. Tests can also have unintended consequences [4,8,9]: CT scans may lead to the diagnosis and treatment of other conditions that would not have caused harm if left undetected; they involve exposure to relatively high levels of radiation [10,11], and there is potential for adverse psychological effects from receiving a positive CAC score [12,13]. These potential harms need to be balanced with potential benefits for individual patients, which could include avoiding lifelong medication for patients with an intermediate traditional risk score, who are reclassified to low risk based on CAC, or earlier access to effective preventative medications in patients reclassified to high risk. In addition, the opportunity to visualise coronary plaque with their physician can be a personalised motivational tool that is associated with enhanced adherence to preventative strategies [14].

Shared decision-making is essential for the primary prevention of CVD, where the benefits and harms of prevention medication are highly dependent on individual values [15]. People need support to understand their risk of a heart attack or stroke and to weigh the benefits and harms of prevention options, which may not reflect arbitrary medication thresholds as recommended in professional guidelines [16]. Previous research has found significant variation in people’s attitudes about the benefits of medication when presented with different CVD risk information [17]. Decision aids are effective tools to implement shared decision-making when there is not one clear option by helping people understand the clinical evidence and consider how this relates to their values [18]. In the CVD prevention context, decision aids have been shown to improve uptake and self-reported adherence to preventive interventions [19] and to improve patients’ understanding of their risk of having a CVD event [20], including those with lower health literacy [21].

The addition of CAC scores into CVD risk assessment pathways in various guidelines around the world [2,22] presents new communication challenges. Health literacy is associated with higher levels of CVD risk factors and worse health outcomes. Limited health literacy has been consistently associated with poorer overall health status and all-cause mortality, greater use of emergency care, lower screening utilisation, poorer ability to interpret health messages, poorer ability to demonstrate appropriate use of medicines, increased lifestyle-related risk factors, and poorer self-management of chronic conditions [23,24,25,26,27,28]. Health literacy may also affect primary CVD risk management decisions, with people with lower health literacy potentially more willing to accept medication as first-line management and less motivated and willing to make recommended lifestyle changes [29]. Health literacy needs can be met using evidence-based methods to present probabilities about CVD risk, the benefits and harms of tests and prevention options, and associated uncertainties [30]. Prior reviews show that existing CVD prevention tools and cardiology procedure consent forms [31] do not meet international standards for patient decision aids [32] or universal precaution principles for health-literate design [33,34].

This study aims to identify and review the quality of existing patient decision aids and educational materials that may support decision-making about CAC for primary CVD prevention (i.e., for people with no prior history of CVD) against international standards for patient decision aids and health literacy criteria, including readability, understandability, and actionability.

## 2. Materials and Method

### 2.1. Study Design

We used an environmental scan methodology [35], adapted from previous studies [33,36,37]. The environmental scan involves systematically searching publicly accessible, health consumer-facing websites in order to capture resources that consumers may find in a search for health information, as opposed to a systematic search of the academic literature, which does not have the scope to capture these resources.

### 2.2. Inclusion and Exclusion Criteria

Eligible tools: (1) provided information about CAC scans; (2) were aimed at the public rather than clinicians, (3) focused on primary CVD prevention (i.e., not secondary prevention or treatment of established CVD), (4) were freely available at the time of the review, and (5) were written in English. Exclusion criteria included: (1) material that could not be viewed due to technical problems after two attempts on two different days, (2) material aimed at health professionals or clinicians (including academic literature); (3) material that required a login or payment to access, and (4) material that was a media release or news item.

### 2.3. Search Strategy

We used two main strategies to identify websites containing eligible patient resources: 1. a manual search of known decision aid repositories and heart health/cardiology organisations (Appendix A); 2. a systematic search using Google Australia. The search terms were agreed upon with input from authors with expertise in CVD prevention, medical decision-making, clinical epidemiology, and primary care. Twelve search terms were used to capture resources about CAC scans (CAC, CACS, calcium heart score, calcium score screening, computerised tomography calcium score, coronary artery calcium scoring, coronary calcium scan, coronary calcium score, coronary calcium test, coronary CT calcium score, CT calcium score, CTCS). Each of these terms was combined with the term “decision aid” (e.g., “CAC” AND “decision aid”), resulting in 12 unique searches. The first 50 results from each search were considered, as previous research has indicated minimal new relevant results after this [34].

#### 2.3.1. Known Repository Search

Two authors (SM, NF) manually searched the compiled websites using the search terms described above, capturing decision aids and patient education materials. Two further websites were identified during the search (Appendix A). The first 50 results in each search were considered for inclusion.

#### 2.3.2. Google Search

Two authors (SM, SC) conducted searches using Google.com.au (accessed on 11 October 2021). Each reviewer used an incognito browser window and reset the cache in their browser windows before each search to minimize the effect of Google search optimization. The first 50 results from each search were exported to Excel using the SEOQuake browser extension, with some searches yielding fewer than 50 results. Eleven of the twelve search term combinations yielded the same number of results from both reviewers. The one remainder search term combination was redone on 18 October 2021, due to a large discrepancy in total items returned (50 to 11). Both reviewers returned 15 results the second time. The results from both searches for this search term combination were included in the total pool of URLs.

Results from both search strategies (known repository and Google search) were combined, and duplicate results were removed. All results were screened independently for inclusion by two authors. Discrepancies (*n* = 8) were decided by consensus with the involvement of a third author. An additional resource was added at this stage, as a link embedded in a result that was excluded due to lack of direct relevance to the topic was found to lead to an eligible patient education resource that was not found by either search strategy.

### 2.4. Data Extraction and Evaluation

#### 2.4.1. Decision aid Qualification

Two authors (CBatcup, SC) used a predefined data extraction form (Appendix A) to record basic descriptive information on each eligible resource (e.g., source organisation, country of origin, publication date) to assess whether the resource qualified as a decision aid according to the International Patient Decision Aid Standards Inventory version 4 (IPDASi) [32] and to record information provided in each resource about the benefits and harms of CAC (including estimates of effect). The IPDASi qualifying criteria consist of seven items evaluated on a Yes/No scale (Box 1). All criteria must be met for a resource to qualify as a decision aid.

Box 1IPDASi v4 qualifying criteria.The decision aid describes the condition (health or other) related to the decision.The decision aid describes the decision that needs to be considered.The decision aid identifies the target audience.The decision aid lists the options (health care or other).The decision aid has information about the positive features of the options (e.g., bene-fits, advantages).The decision aid has information about the negative features of the options (e.g., harms, side effects, disadvantages).The decision aid helps patients clarify their values for outcomes of options by: a) asking people to think about which positive and negative features of the options matter most to them AND/OR b) describing each option to help patients imagine the physical, social, and/or psychological effect.

#### 2.4.2. Understandability and Actionability

Two authors (CBatcup, NF) independently assessed the resources against the Patient Education Material Evaluation Tool for Printable Materials (PEMAT-P) [38], a validated tool to assess the understandability and actionability of patient information materials. The PEMAT-P consists of two percentage scores: (1) understandability (measure of how well a person can process and explain the key message of the material; a 17-item subscale), and (2) actionability (a measure of how well a person is able to identify what to do based on the information presented; a seven-item subscale). Items are rated on a Yes/No scale or Not Applicable for certain items. Higher percentage scores indicate better understandability and actionability. While the PEMAT-P does not have empirical thresholds for acceptable levels of understandability and actionability, the developers of the PEMAT-P have previously set a threshold of 70% for material to be considered understandable and actionable, with materials that score at or below 70% considered to be poorly understandable and actionable [38]. Any discrepancies in rating were resolved by consensus.

#### 2.4.3. Readability

We used the Sydney Health Literacy Lab Health Literacy Editor (SHeLL Editor) [39] to assess several measures of readability. The Simple Measure of Gobbledygook (SMOG) [40] was used to assess grade reading level. Materials written up to a grade 8 reading level (inclusive) are considered to be suitable for most users, although some agencies recommend that materials be written to grade 5 or 6 levels to ensure the widest understanding [41]. The SHeLL Editor also measures the percentage of complex language in written materials, the number of times the passive voice is used, and lexical density using the Measure of Lexical Textual Diversity (MLTD) [42]. Lexical density score indicates how close written text is to spoken English. Spoken English generally has a lower lexical density score (1.5–2), while written English generally has a higher score (3–6) [43]. A lexical density score below three is recommended for written health information [44].

## 3. Results

### 3.1. Search Results

The Google search resulted in two independent pools of 519 and 480 URLs, with a total of 460 after the removal of duplicate URLs. The search of known repositories yielded 12 URLs after the removal of duplicates. After applying inclusion and exclusion criteria, 13 items were found to meet the criteria for inclusion, of which nine were identified by Google search and three via a search of known repositories and websites. One additional item was identified within an excluded Google search result. Reasons for exclusion are provided in Figure 1. Appendix A provides the list of all included online resources. The characteristics of the included resources are summarised in Table 1.

### 3.2. Name of Test

There was marked variability between the results as to the name given to the test, with four items using more than one name within the one resource. ‘Coronary calcium scan’ was the most frequently used name (*n* = 5), with ‘calcium scan,’ ‘coronary artery calcium scan,’ and ‘coronary artery calcium scoring’ each used in two items. Other names used included ‘heart scan’ and ‘heart test,’ ‘cardiac computed tomography (CT) for calcium scoring,’ ‘CT scan screening,’ ‘calcium score screening heart scan,’ and ‘coronary calcium score test.’

### 3.3. Evaluation of Online Resources

One (8%) online resource met all seven IPDASi criteria to qualify as a decision aid (Table 2; see Appendix A for the evaluation of each resource). Six resources (46%) provided information about the potential benefits and harms of CAC scans. Two (15%) only provided information about benefits, while one (8%) only provided information about harms. Five (38%) provided general information about CAC but did not include any information about outcomes (harms or benefits). Potential benefits of undergoing a CAC scan that were described by the resources included that the test: is convenient and non-invasive with no side effects; facilitates early detection and treatment of disease; provides more information about one’s individual CVD risk, which can be used to guide treatment decisions, especially for those at moderate or uncertain risk of CVD; and it can prompt/motivate lifestyle change or can help those with low risk avoid unnecessary medications. The most commonly mentioned potential benefit was that the test provides more information to guide decision-making. Potential harms that were described included exposure to radiation, cost of the scan, that risk may be over- or underestimated, and that the scan could lead to unnecessary tests and treatment. Radiation exposure was the most commonly mentioned potential harm. Only one (8%) resource identified the potential benefits and harms of not undergoing a CAC scan. The potential benefits described for this were: avoiding paying for unnecessary tests and avoiding unnecessary treatment or worry. The potential harms described were not having all the information needed to reduce risk (for people at medium risk of CVD on the basis of an absolute risk score).

Only one item included quantitative information about potential harm, stating that a CT scanner emits approximately the same radiation as 10 X-rays from an X-ray machine. None of the resources included any information on estimates of effect for both benefits and harms.

On average, the resources scored poorly for understandability and actionability on the PEMAT-P scale, although there was considerable variability, with only three items scoring >70% for both understandability and actionability. The mean understandability score was 68% (range 21% to 94%) and mean actionability score was 48% (range 0% to 100%). The average reading grade level was grade 12.8. None of the resources were written to the recommended grade eight level. Appendix A includes the readability scores for all resources.

## 4. Discussion

None of the thirteen eligible materials were suitable for both facilitating informed decision-making and catering to patients with varying health literacy needs. One item met all seven IPDASi criteria to qualify as a decision aid and scored > the 70% threshold for both understandability and actionability but did not meet the recommended grade reading or lexical density score. The mean understandability and actionability scores (68% and 48%, respectively) can be compared to our previous findings about patient education materials and decision aids for related conditions. On average, the materials in this study have poor understandability and actionability, scoring slightly higher than the mean understandability score for online CVD risk calculators (64%/19%) [34], hospital consent forms for cardiology procedures (62%-understandability only) [31], and online heart failure information (56%/35%) [45], but considerably lower than online decision aids for primary CVD prevention (87%/61%) [33]. In line with our findings in this current study, reading grade levels of decision aids and other patient materials in comparable studies were also consistently higher than recommended [33,34,45], demonstrating a widespread failure to comply with the principles of universal health literacy precautions [46]. It was also notable that across the thirteen items, there were ten different names given for CAC scans. Multiple and confusing names for one test could affect patients’ ability to understand information about the test and their medical care. This variability cannot simply be explained by differences in what CAC scans are called in different countries, as 12/13 items were developed in the USA.

Clinical guidelines that include CAC scans for primary prevention need to be supported with better patient education materials and risk communication tools to ensure that patients can participate in an informed, shared decision-making process about CVD prevention. This could take the form of a decision aid that is informed by universal precautions for health literate design [46] and explains the quantitative benefits and harms of CAC scans, using best practices for communicating risk and quantitative information [30]. It is noteworthy that none of the thirteen resources included estimates of effect for both benefits and harms. These might include the number of people correctly and incorrectly reclassified as high risk or low risk when CAC score is added to the traditional risk score [7].

This study has documented the current lack of availability of appropriate tools to support patient decision-making around CAC scans for CVD screening. This is relevant beyond CAC, as emerging diagnostic tools such as polygenic risk scoring and blood-based multi-omic signatures come to the clinic [8]. It is critical that evidence-based tools for communicating the potential benefits and harms of such tests and for supporting patient decision-making are developed prior to their widespread adoption in practice. There is some evidence that routine use of CAC scans has already been quite widely adopted in practice, with a recent study showing that they are routinely offered in almost half of executive health screening programs offered by top-ranked cardiology hospitals in the United States, despite the fact that clinical guidelines do not recommend this in asymptomatic adults [47].

The strengths of this study include the use of the rigorous environmental scan methodology, which uses a systematic search method to describe the actual landscape of online resources available to health consumers outside of the scope of a traditional review of the academic literature. All steps of the systematic search, data extraction and data analysis were performed by two independent reviewers.

Limitations include the fact that Google search results are dynamic and location dependent and are not likely to be replicable. Additionally, there was one Google search combination where the independent reviewers returned a discrepant number of results, probably due to a search syntax error. The search was reperformed at a later date with accordant results, and the results of both searches were included in the total number of URLs to maximise comprehensiveness, which would further limit the replicability of the search. Our search only included English language results, which potentially excluded relevant online resources in other languages. Our search was specifically designed to find publicly available materials that may qualify as a patient decision aid and thus will have excluded patient education materials about CAC scans that are not intended to support decision-making. This study aimed to identify materials that may be found by consumers using Google or general consumer websites relating to decision aids or heart disease. Specialist organisations (e.g., the Society of Cardiovascular CT) also produce patient education materials but were not specifically included in our search method, as such websites are unlikely to be targeted by consumers without heart disease who are seeking information about prevention. Such organisations may need to optimize their website analytics if they want consumers to find their materials more easily. Our search may have also excluded materials used in health facilities that are not available online.

Future research could explore how to explain uncertainties around using CAC to inform primary prevention decisions, as this has been identified as a major gap in the risk communication literature [30]. For example, we do not know how best to explain the mixed evidence in the literature and the strength of this evidence overall when providing estimates of outcomes. However, guidelines exist to support the development of high-quality decision aids that address varying health literacy needs. Better tools are needed to support shared decision-making for CAC where it is already used in clinical practice. This needs to be informed by clinical guidelines to ensure local relevance, highlighting the importance of potential harms, benefits and patient concerns, and integrated into clinical workflows to facilitate best practice communication about this issue.

## 5. Conclusions

Current decision support materials available to consumers considering coronary artery calcium scoring do not meet the criteria to enable informed decision-making and do not meet the health literacy needs of the general population. This echoes similar issues identified within cardiology procedure consent forms [31]. Clinical guidelines that include CAC scans for primary prevention must be supported by best-practice decision aids to support patient decision-making about this issue.

## Figures and Tables

**Figure 1 ijerph-19-11705-f001:**
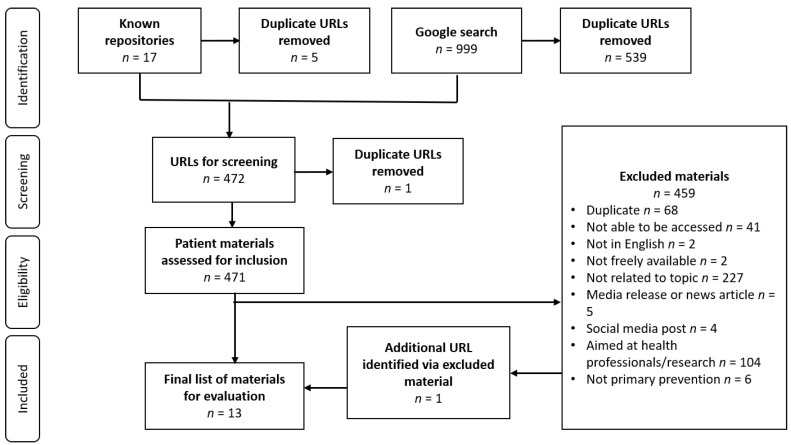
Study search strategy and results.

**Table 1 ijerph-19-11705-t001:** Characteristics of included online resources.

Characteristic	*n* (%)
Format	
Interactive	1 (8)
Static	12 (92)
Country of Origin	
Australia	1 (8)
United States	12 (92)
Year of publication or last update	
2016–2021	6 (46)
2015 or earlier	1 (8)
Not stated	6 (46)

**Table 2 ijerph-19-11705-t002:** Evaluation of online resources.

Evaluation Criteria	Mean	Range
IPDASi * qualification [32]	3.5	0–7
PEMAT-P ^†^ [38]		
Understandability	68%	21–94%
Actionability	48%	0–100%
Readability		
Grade reading level (SMOG ^‡^) [40]	12.8	10–15
Complex language (%)	24.4	19.7–31.4
Passive voice (# of instances)	12.2	4–22
Lexical density (MTLD ^#^) [42]	4.2	3.3–5.4

* International Patient Decision Aid Standards; ^†^ Patient Educational Material Assessment Tool for Printed materials (70% is the upper threshold for understandability and actionability); ^‡^ Simple Measure of Gobbledygook (grade 8 level and below recommended); ^#^ Measure of Lexical Textual Diversity (score below 3 recommended).

## Data Availability

Website URLs containing the decision support tools are available in Appendix A. Descriptive and evaluative data are also available within Appendix A.

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
