# Peer review of "Decision Support Tools for Coronary Artery Calcium Scoring in the Primary Prevention of Cardiovascular Disease Do Not Meet Health Literacy Needs: A Systematic Environmental Scan and Evaluation"

_ijerph, 2022, doi:10.3390/ijerph191811705_

Round 1

Reviewer 1 Report

This study has significant methodological limitations.  The most reliable source of conclusions at the moment is a systematic review and a meta-analysis with a very detailed analysis of published data and available scientific evidence and limitations of the methods studied, which also allow conclusions to be drawn about the use of these techniques.  The presented study contains the necessary information, and the data presented in it are very limited and may not be representative due to incorrect methodology.

Author Response

This study has significant methodological limitations.  The most reliable source of conclusions at the moment is a systematic review and a meta-analysis with a very detailed analysis of published data and available scientific evidence and limitations of the methods studied, which also allow conclusions to be drawn about the use of these techniques.  The presented study contains the necessary information, and the data presented in it are very limited and may not be representative due to incorrect methodology.

Response: We thank the reviewer for the time they have taken to read our study. However, the aim of our review was not to assess effectiveness in academic papers; it was to document the quality of publicly available education materials. We therefore do not have the data required for a meta-analysis. An environmental scan is the correct method for this aim and it has been used in numerous health topics in previous research (see references). We have added some extra information to the Methods section describing our study design, to clarify how our method addresses our aim.

Revised text (p5): The environmental scan involves systematically searching publicly accessible, health consumer-facing websites, in order to capture resources that consumers may find in a search for health information, as opposed to a systematic search of the academic literature which does not have the scope to capture these resources.

Reviewer 2 Report

The manuscript deals with rare and challenging topic – analysis of existing patient decision aids and education materials about the coronary calcium scoring (CAC) about CAC for primary prevention of cardiovascular diseases.

There are plenty of publications about CAC in the international literature. But vast majority of them are directed to medical specialists, not to the patients. So far no one national or international cardiological society recommends CAC as a primary tool for screening for coronary artery disease. As indicated in the manuscript, CAC is usually regarded as an axillary tool for CVD risk stratification and selection of most appropriate management of the patient according to the defined CVD risk (use of statins).

The manuscript has all necessary parts.

In general, results of the reviewed study are quite similar to more general publication in BMJ (ref.33): Authors of BMJ paper found before that most of online decision aids for primary CVD prevention had limitations and were in poor compliance with International Patient Decision Aids Standards (IPDAS). Usually they have good understandability and aveage actionability.

The most important positive message resulting from the reviewed study is the need for creation of better patient education materials and decision aids about advantages and possible dangers of CAC.

There some remarks and comments to the authors of the paper:

-          The title of the manuscript is: Decision support tools for coronary artery calcium scoring in the primary prevention of cardiovascular disease do not meet  health literacy needs: a systematic environmental scan and  evaluation”. However, it looks that it does not fully correspond with results of the study. Authors found good a good understandability of CAC patient decision tools (68%) and mediocre actionability (48%). Such title creates in general somewhat negative impression about CAC, which is a well-established imaging modality, backed up with multiple scientific publications. There are no doubts about its value for management of patients when it is used according to approved indications.

-          International Medical community is just begins to create web-based decision support tools according to some national or international standards. Most of reviewed education materials for patients regarding CAC has been created by different societies and organizations for information purposes, not as decision aids. So far majority of these organizations do not regard International Patient Decision Aids Standards (IPDAS) as an obligatory standard for production of patient-oriented materials. In this case, an attempt to check the adherence of these materials to IPADS looks a little bit artificial.

-          Appendix A just gives a list of websites of heart/cardiology organizations, but not references to CAC educational materials.

-          Authors missed to include in their analysis the materials from the Society of Cardiovascular CT (SCCT). SCCT is well known internationally for creation of high-quality guidelines and statements on cardiac CT, including CAC. Their web-page has information for patients (https://scct.org/page/For_patients). 

Author Response

The manuscript deals with rare and challenging topic – analysis of existing patient decision aids and education materials about the coronary calcium scoring (CAC) about CAC for primary prevention of cardiovascular diseases.

There are plenty of publications about CAC in the international literature. But vast majority of them are directed to medical specialists, not to the patients. So far no one national or international cardiological society recommends CAC as a primary tool for screening for coronary artery disease. As indicated in the manuscript, CAC is usually regarded as an axillary tool for CVD risk stratification and selection of most appropriate management of the patient according to the defined CVD risk (use of statins).

The manuscript has all necessary parts.

In general, results of the reviewed study are quite similar to more general publication in BMJ (ref.33): Authors of BMJ paper found before that most of online decision aids for primary CVD prevention had limitations and were in poor compliance with International Patient Decision Aids Standards (IPDAS). Usually they have good understandability and aveage actionability.

Response: Thank you for your positive comments about the manuscript. In regards to similarities between this study and the BMJ Open paper in question, we would point out that that study looked at online decision aids for primary CVD prevention, and did not consider patient education materials for patients undergoing CAC scoring.

The most important positive message resulting from the reviewed study is the need for creation of better patient education materials and decision aids about advantages and possible dangers of CAC.

There some remarks and comments to the authors of the paper:

  1. The title of the manuscript is: Decision support tools for coronary artery calcium scoring in the primary prevention of cardiovascular disease do not meet  health literacy needs: a systematic environmental scan and  evaluation”. However, it looks that it does not fully correspond with results of the study. Authors found good a good understandability of CAC patient decision tools (68%) and mediocre actionability (48%). Such title creates in general somewhat negative impression about CAC, which is a well-established imaging modality, backed up with multiple scientific publications. There are no doubts about its value for management of patients when it is used according to approved indications.

Response: The title “Decision support tools for CACS in the primary prevention of CVD do not meet health literacy needs” clearly refers to decision support tools, and makes no comment on the clinical or scientific validity of CAC scoring. As mentioned in the section Methods: Understandability and Actionability, the established PEMAT-P threshold for understandability and actionability is 70%. Materials that score below 70% are considered to have poor understandability and actionability. Additionally, the materials on average scored poorly in terms of the recommendations for readability of health information. As such, we stand by the statement that the decision support tools evaluated in this study do not meet health literacy needs.

  1. International Medical community is just begins to create web-based decision support tools according to some national or international standards. Most of reviewed education materials for patients regarding CAC has been created by different societies and organizations for information purposes, not as decision aids. So far majority of these organizations do not regard International Patient Decision Aids Standards (IPDAS) as an obligatory standard for production of patient-oriented materials. In this case, an attempt to check the adherence of these materials to IPADS looks a little bit artificial.

Response: While it is true that the majority of organisations does not regard IPDAS as an obligatory standard, the first version of the IPDAS criteria were published in 2006, so the standards are not new. This study only uses the IPDAS criteria to determine whether a material qualifies as a decision aid or not. The materials were not evaluated according to the criteria determining their quality as a decision aid. Therefore, we do not think it is artificial or inappropriate to use the IPDAS criteria in this circumstance.

  1. Appendix A just gives a list of websites of heart/cardiology organizations, but not references to CAC educational materials.

Response: Appendix A is not intended to reference the CAC education materials that were included in this study. Appendix A contains a table titled “List of known decision aid repositories and heart health/cardiology organization websites” and is provided to give details about our search strategy (see manuscript section 2.3). The list of included education materials is found in Appendix C, Table C.1.

  1. Authors missed to include in their analysis the materials from the Society of Cardiovascular CT (SCCT). SCCT is well known internationally for creation of high-quality guidelines and statements on cardiac CT, including CAC. Their web-page has information for patients (https://scct.org/page/For_patients). 

Response: Our results only include materials that were produced from our search strategy, which included google search terms and general decision aid or heart disease related organizations that target consumers. The Society of Cardiovascular CT membership is targeted at physicians and technical experts, so unlikely to be known to consumers. However we acknowledge that it does produce patient education materials that could be useful. We have therefore included this as an additional point in the discussion.

Revised text (p14-15): This study aimed to identify materials that may be found by consumers using Google or general consumer websites relating to decision aids or heart disease. Specialist organisations (e.g. Society of Cardiovascular CT) also produce patient education materials but were not specifically included in our search method, as such websites are unlikely to be targeted by consumers without heart disease who are seeking information about prevention. Such organisations may need to optimize their website analytics if they want consumers to find their materials more easily.

Reviewer 3 Report

The authors present an excellent and unique analysis of educational materials related to CAC scoring. Their conclusion is a dearth of quality and readable materials pertaining to this pervasive and ever growing diagnostic test, which is extremely important given the test's importance in clinical management of patients. 

Author Response

The authors present an excellent and unique analysis of educational materials related to CAC scoring. Their conclusion is a dearth of quality and readable materials pertaining to this pervasive and ever growing diagnostic test, which is extremely important given the test's importance in clinical management of patients. 

Response: We thank the reviewer for their kind comments about our study.